# Oxidation Behaviour of Microstructurally Highly Metastable Ag-La Alloy

**DOI:** 10.3390/ma15062295

**Published:** 2022-03-20

**Authors:** Andraž Jug, Mihael Brunčko, Rebeka Rudolf, Ivan Anžel

**Affiliations:** Faculty of Mechanical Engineering, University of Maribor, Smetanova ulica 17, 2000 Maribor, Slovenia; andraz.jug1@student.um.si (A.J.); mihael.bruncko@um.si (M.B.); ivan.anzel@um.si (I.A.)

**Keywords:** Ag-La alloy, rapid solidification, metastable microstructure, internal oxidation, characterisation, formation mechanism

## Abstract

A new silver-based alloy with 2 wt.% of lanthanum (La) was studied as a potential candidate for electric contact material. The alloy was prepared by rapid solidification, performed by the melt spinning technique. Microstructural examination of the rapidly solidified ribbons revealed very fine grains of α_Ag_ and intermetallic Ag_5_La particles, which appear in the volume of the grains, as well as on the grain boundaries. Rapid solidification enabled high microstructural refinement and provided a suitable starting microstructure for the subsequent internal oxidation, resulting in fine submicron-sized La_2_O_3_ oxide nanoparticle formation throughout the volume of the silver matrix (α_Ag_). The resulting nanostructured Ag-La_2_O_3_ microstructure was characterised by high-resolution FESEM and STEM, both equipped with EDX. High-temperature internal oxidation of the rapidly solidified ribbons essentially changed the microstructure. Mostly homogeneously dispersed nano-sized La_2_O_3_ were formed within the grains, as well as on the grain boundaries. Three mechanisms of internal oxidation were identified: (i) the oxidation of La from the solid solution; (ii) partial dissolution of finer Ag_5_La particles before the internal oxidation front and oxidation of La from the solid solution; and (iii) direct oxidation of coarser Ag_5_La intermetallic particles.

## 1. Introduction

In practice, nearly all metallic materials are thermodynamically unstable, and convert to oxides in most high-temperature environments. The proceeding of the oxidation process depends on the materials’ composition and on the environmental conditions (temperature and oxygen partial pressure) [1,2,3]. Basically, the high-temperature oxidation of alloys can be divided into two limiting cases: (i) oxidation processes where only external oxide phases are formed; (ii) oxidation processes with internal oxide formation, which can be terminated by transition to the external oxidation. In the first case, the reaction begins at the metal/gas interface, and the reaction products form an intermediate layer between the alloy and the gas. In the second case, the high-temperature oxidation starts with the dissolution of oxygen into the lattice of the metal matrix, and continues with diffusion of the oxygen atoms into the volume of the matrix, where a selective reaction with the less noble solute or second phase occurs—internal oxidation (IO). Both modes of high-temperature oxidation are mostly undesirable, while they often cause corrosive failure of engineering components. On the other hand, controlled internal or external oxidation can be used for improving the properties and corrosion resistance of metallic materials [4,5,6].

Silver (Ag) has the highest electrical and thermal conductivity of all today’s known metals, and is therefore an excellent material for electric contacts [7,8,9]. However, pure Ag does not satisfy all the criteria for an efficient contact material, and somehow needs improvement to be useful in industrial applications. The main weaknesses include poor mechanical properties and insufficient resistance of the surface against an electric arc [10,11,12]. Nowadays, it is already well known that fine dispersed nanosized oxide particles in an Ag-matrix can be successfully used for improving the aforementioned shortcomings [13,14,15,16]. In comparison to solute atoms and precipitates, the fine oxide particles are much more effective as obstacles for dislocation movement at high temperatures [17]. To retain the appropriate mechanical properties at high temperature, at least 4 vol.% of non-shareable, incoherent oxide particles with a high crystallographic mismatch at the matrix–oxide interface is necessary [18]. Namely, the strengthening in this case is associated with relaxation of the dislocation core stresses at the interface during climbing and the additional force, which is, consequently, required for detachment of the dislocation from the oxide particle [18,19]. This mechanism is active mostly at higher temperatures, where dislocation climbing cannot be neglected. The fine dispersion of oxide particles also improves the functional properties of the contact materials. Discharged energy during the arc causes melting of the matrix, and also the decomposition of the oxides. In the process of decomposition, arc energy is absorbed by the oxides, which accelerates its extinguishing/quenching. It was found that the size of oxide particles has a great influence on the decomposition velocity. Smaller oxide particles are considered to sublimate more easily, and, consequently, they cause more rapid arc discharge [12,20]. On the other hand, the oxide particles should not vaporise and sublimate rapidly during arc discharging. Namely, they need to increase the viscosity of the melt and prevent matrix erosion through splattering. While floating in the melt, the oxide particles need to be distributed uniformly within it, to prevent the formation of the oxide layer on the contact surface when it solidifies [21].

One of the most promising methods for attaining the uniform fine dispersion of oxides in the metal matrix is high temperature IO. In this process, oxygen from the surrounding atmosphere is adsorbed onto the metal surface, dissolves in the metal crystal lattice, and diffuses inwards. During diffusion through the crystal lattice, the oxygen atoms react with the dissolved atoms of the less noble solute element, and form the oxides that precipitate from the solid solution after the solubility product for the oxide is exceeded. Alternatively, in the case of the alloying systems with the insoluble less noble alloying element, the oxygen atoms, diffusing to the interior of the alloy, react with the second phase particles, that include the atoms of the reactive alloying element. In this case, the particles of the internal oxide keep the same spatial distribution and the same size and shape as the second-phase particles originally present in the alloy. In this type of so-called direct oxidation of the second phase particles, fine dispersion of the nanosized oxides can be attained only when the second phase particles are of small size, and also dispersed homogeneously through the volume of the alloy. Conversely, if the diffusivity of the alloying element in the matrix is higher and the second-phase particles are small enough and finely dispersed, so that the solubility of the alloying element in the matrix is also higher (the equilibrium concentration of the alloying element in the matrix increases with increasing curvature of the second-phase particles), practically all the atoms of the alloying element are oxidised from the solid solution. In fact, the oxide-forming alloying element is provided by the dissolution of the finely dispersed second-phase particles ahead of the oxidation front and diffusion of the alloying atoms towards the reaction interface [6]. This type of IO (diffusional internal oxidation) makes it possible that the whole amount of the low-soluble alloying element oxidises from the solid solution, and produces a uniform distribution of very fine oxide particles in the matrix. Therefore, it is clear that the success of producing suitable oxide particles in two-phase alloys depends on the initial microstructure and the IO conditions.

In the past, Cd-oxides have been used in the Ag-based contact materials to improve their mechanical and electrical properties. Although the switching performances of these materials are still known as the greatest, Cd-oxide has been replaced gradually by Sn-oxide, due to toxic vapours’ release into the environment while operating [21,22,23,24]. However, very soon it became obvious that Sn-oxide does not satisfy all the considered criteria either [25]. Numerous oxides have been investigated since then, and the oxides of rare earth elements have also been recognised as promising candidates [25,26]. One of them, namely La-oxide, has already been identified as a highly stable compound which does not decompose easily, and, therefore, increases the viscosity of molten silver and reduces the splash erosion [15,27].

The Ag-La alloying system is characterised by the extremely low solubility of lanthanum in silver. Namely, the maximum value of only 0.06 wt.% of La is attained at a eutectic temperature [28]. Consequently, the solidification microstructures of Ag-based alloys alloyed by La are always two-phase, even in the cases of microalloying. The microstructure is composed of α_Ag_ primary dendrites and coarse lamellar eutectic (α_Ag_ + Ag_5_La). Therefore, it is not possible to achieve fine dispersion of intermetallic Ag_5_La particles in the Ag-matrix by conventional casting methods. Consequently, according to the known mechanisms of IO in two phase alloys [29], it is also not possible to expect to reach the nanosized finely dispersed La-oxides during the subsequent IO of conventionally cast silver alloy with a coarse Ag_5_La intermetallic. On the other hand, rapid solidification (RS) has often been used successfully as a means for refinement of the microstructure and to decrease the size of phases [29,30]. Besides this, increasing solidification rates lead to metastable microstructures [16,30,31] (increasing the solubility of the alloying elements for instance), which could enable at the subsequent IO obtaining fine, nanosized oxide particles. Therefore, RS performed by the melt spinning technique was used in our research work to investigate the influence of the starting highly metastable microstructure in the selected Ag—2 wt.% La alloy on the IO proceedings, and the process tracing served as a tool to define the mechanism of oxidation. Additionally, high-temperature IO in hypereutectic Ag—14 wt.% La was studied in the as-cast state of the alloy, to clarify the oxidation mechanism of the Ag_5_La intermetallic phase. The objective was to identify an optimal microstructure that will lead to uniformly distributed, nanosized, incoherent oxide particles in the Ag matrix.

## 2. Materials and Methods

The experimental Ag-La alloys with 2 and 14 wt.% La were prepared by induction vacuum melting using high purity elements (Ag—99.99 wt.%; La—99.9 wt.%) as the starting materials. Chemical composition was confirmed by X-ray Fluorescence (XRF) (Thermo Scientific, Waltham, MA, USA), as well as Inductively Coupled Plasma–Optical Emission Spectrometry (ICP-OES) (HP, Agilent 7500 CE, Santa Clara, CA, USA).

Rapidly solidified ribbons were prepared by the Chill-Block Melt Spinning technique, which offers not only the benefits of rapid solidification, but also facilitates the study of the mechanism and kinetics of the IO process. Particularly, the rectangular cross-section of the rapidly solidified ribbons with high ratio between width and thickness enables the assumption of the one-dimensional diffusion of oxygen and lanthanum during the internal oxidation process. Samples of the alloy Ag-2%La weighing about 100 g were remelted in an induction furnace of the Melt Spinning device in a graphite crucible of 26 mm inner diameter under an Ar atmosphere. After that, the melt was ejected through the nozzle (φ = 0.8 mm) of the crucible by an Ar overpressure of 0.02 to 0.03 MPa onto the surface of a Copper-Beryllium wheel rotating at a speed of about 24 m/s. Continuous ribbons were obtained, about 1 mm in width and 50 to 70 µm in thickness.

The experiments of high-temperature IO were performed in a tube furnace consisting of two heating zones and a controlled atmosphere provided by a gas supplying system. In the first zone, which was kept at a much higher temperature than the second one, samples were heated in a very short time to the internal oxidation temperature. After that, the samples were pushed into the second zone of the furnace and annealed there for different times. With this procedure the heating time was shortened, and IO during this time could be neglected. About 3-cm-long samples of rapidly solidified ribbons (with 2 wt.% La), as well as the samples in the as-cast state (14 wt.% La) bulk samples (10 × 10 × 3 mm), were oxidised internally at 973 K in an O_2_ atmosphere for different periods of time. Before the heat treatment, the free surfaces of the samples were cleaned by 65% HNO_3_. The temperature during IO treatment was controlled with a type-K thermocouple placed directly above the samples.

Microstructural characterisation of the samples was carried out using scanning electron microscopes (FEI SEM-Sirion 400 NC) ((FEI, USA), FE-SEM Jeol JSM-7900F, (Jeol, Japan) and a high-resolution scanning transmission electron microscope (STEM Jeol JEM ARM 200F, Jeol, Japan). All microscopes were equipped with EDX detectors (INCA 350, Oxford Instruments, UK), while the FE-SEM Jeol JSM-7900F was additionally attached with an Electron Back Scatter Diffraction (EBSD) detector (Jeol, USA). The cross-sections of the RS ribbons were prepared by a high-performance Cross-Section Polisher, with the added function of specimen cooling and prevention of exposure to the air. The specimens for TEM were prepared by collecting thin ribbons together and moulding them into resin. After that, the specimens were ground in the middle to produce a few nm thick regions in a perpendicular direction. In the case of bulk samples, an optical microscopy Nikon Epiphot 300 (Nikon, Japan) was used mostly to examine the microstructure evolution during IO. The cross-section of the bulk samples was prepared by standard metallographic methods with grinding and polishing.

## 3. Results

### 3.1. Rapidly Solidified Ribbons of the Experimental Ag-La Alloy with 2 wt.% La

Because of their very low thicknesses, the ribbons exhibited a high ratio between the width and the thickness, with practically only two free surfaces. On the macroscopic scale, the lower surface that was in contact with the wheel (Figure 1a) exhibited some dimples, which were formed by gas picked up at the back edge of the melt puddle on the wheel surface and the upper surface of the ribbons (Figure 1b), which was freely exposed to the atmosphere, and was wavy but smooth, without visible defects at higher magnification.

Figure 2a shows the grain size distribution in a transverse cross-section of the rapidly solidified ribbons. Since the main mechanism responsible for faster diffusivity in rapidly solidified ribbons is grain boundary diffusion, the size of grains influences the internal oxidation significantly. The grain size analysis was carried out using the Inverse Pole Figure (IPF) with EBSD mapping. It can be seen that the fine equiaxed grains appeared near the wheel’s surface, and the greater equiaxed grains were spread from the middle of the ribbon to the free surface. According to this, the cross-section microstructure can be divided into two zones: The zone of Fine Equiaxed Grains (FEG) and the zone of Coarse Equiaxed Grains (CEG). The diameter of the grains in the FEG zone extended from 150 nm to 900 nm, while the grains in the CEG zone were in the range from 1 µm to 5 µm. Grains with the size between 700 nm and 3.2 µm represent the major area fraction (>0.07), where the highest area fraction belongs to the grains with the size of 2.1 µm, as shown by the graph in Figure 2b. The formation of only the equiaxed zone without columnar grains can be explained on the basis of the excellent thermal conductivity of the alloy’s melt, and, consequently, the high undercooling obtained in the melt throughout of the whole cross-section of the ribbon.

Besides the size of the grains, the most important characteristics of the rapidly solidified microstructure in the present study are the size, shape, and distribution of the intermetallic Ag_5_La particles. The analysis of the microstructure has revealed that the intermetallic particles appeared as irregularly shaped particles located at the grain boundaries, and as globularly shaped particles founded mostly inside the α_Ag_ grains, as are present in Figure 3a. In the FEG zone, the irregularly shaped intermetallic Ag_5_La particles were about 300 nm in diameter. Some of them were elongated and measured more than 500 nm in length. The globularly shaped intermetallic Ag_5_La particles inside the grains were much smaller, and their size was up to 100 nm. Additionally, the detailed analysis revealed super fine precipitates inside the α_Ag_ grains (Figure 3b). The precipitates of the size of a few nm were distributed homogeneously within the grains.

The analysis of the intermetallic Ag_5_La particles in the CEG zone again revealed two shapes of the particles, irregular and globular (Figure 4a,b). Their distribution was similar to that in the FEG zone, while their size was slightly bigger. Despite their similar distribution, there were more particles located inside the α_Ag_ grains, as shown in Figure 4. In any case, some of the particles inside the α_Ag_ grains were larger, and exhibited irregular or elongated shape, which could measure up to 500 nm in length (Figure 4a). Otherwise, the size of the irregularly shaped intermetallic Ag_5_La particles on the grain boundaries was up to 700 nm, while the diameter of the globularly shaped intermetallic Ag_5_La particles inside the α_Ag_ grains could be up to 150 nm.

The composition of the intermetallic particles was analysed very carefully by EDX. The content of lanthanum was found to be 17.9 at.%, which is very near to the theoretical composition of the Ag_5_La intermetallic compound from the Ag-La phase diagram [28].

Metallographic examination of the ribbons after IO revealed some differences in the microstructure in the bottom (FEG zone after RS) and the top layers (CEG zone after RS) of the ribbons. In the bottom layer the oxides in the interior of the grains exhibited a bimodal distribution of the particle size: the globular shaped oxides ranging from 20 up to 100 nm (Figure 5a) and the super fine oxides in the order of a few nm (Figure 5b). On the other hand, the oxide particles at the grain boundaries of this layer were elongated, with lengths up to 300 nm (Figure 5a). It was identified that the size of the globular shaped oxide particles inside the grains was much smaller than the size of the intermetallic Ag_5_La particles in the rapidly solidified ribbons prior to IO. Their distribution also corresponded to that of the globular shaped intermetallic Ag_5_La particles. Therefore, the globular shaped oxides in the bottom layer had to be formed by direct oxidation of the Ag_5_La intermetallic compound.

In the top layer, most of the oxide particles are located preferentially along the grain boundaries, as can be seen in Figure 6. They retained an elongated shape with a shorter length (up to 500 nm) in comparison to the length of the irregularly shaped intermetallic Ag_5_La particles before oxidation (up to 700 nm). However, globular-shaped oxide particles (diameter of about 100 nm) were also distributed uniformly throughout the α_Ag_ grains. Therefore, considering the fraction of the particles of different sizes, the mean oxide particle size in the top layer of the ribbons was about 140 nm. In fact, the particles became larger with the increasing distance from the free surface. Namely, the intermetallic particles tend to coarsen before the oxidation front reaches them. Then, the Ostwald ripening stops. Therefore, the oxide particles were smaller closer to the surface than in the interior of the ribbon. Additionally, the comparison of the shape of the particles before and after oxidation has shown that the oxide particles are more round without sharp edges. The size and distribution of the oxide particles in this region can be explained as being a consequence of the direct oxidation of the dispersed intermetallic Ag_5_La particles.

The grain size analysis was also carried out after IO, and is shown in Figure 7. The measured grains’ diameters were in the range from 150 nm to 900 nm in the bottom layer (FEG zone after RS), and between 1 µm and 4.7 µm in the top layer (CEG zone after RS). Grains with sizes between 1.6 µm and 3 µm represent the major area fraction (>0.07), where the highest area fraction belongs to the grains with the size of 2.3 µm. It was found that the size of the grains did not increase significantly. The main reason for this was that the time of the IO process was too short (1 h) for grain growth, and, in addition to this, the intermetallic particles at the grain boundaries inhibited the grain boundary movement.

Microchemical analysis was performed throughout the microstructure of the internally oxidised ribbons. Both the line microchemical analysis shown in Figure 8 and the chemical mapping shown in Figure 9 confirm the lanthanum oxide particles.

Detailed analysis of the oxide particles was performed by STEM. The bright field image (Figure 10a) shows the border region between the α_Ag_ grains and La-oxide, where the boundary between the phases seems to be a few nm wide amorphous zone. It was also found that the coarse particles of La_2_O_3_ were polycrystalline. The Fast-Fourier Transform (FFT) pattern from the oxide phase (Figure 10b) shows the reflections that correspond to the oxide La_2_O_3_.

### 3.2. Hypereutectic Experimental Ag-La Alloy with 14 wt.% La

The cast microstructure of this alloy consists of intermetallic Ag_5_La primary dendrites and eutectic in the interdendritic space, composed of the intermetallic Ag_5_La and α_Ag_ lamellae shown in Figure 11a. The microstructure with coarse primary dendrites, about 100 µm in size, allowed time-dependent monitoring of the oxidation process in the intermetallic Ag_5_La compound. After 6 h of internal oxidation in pure oxygen, the microstructure result was partially oxidised, with about a 260-µm-deep region of internally oxidised zone, as shown in Figure 11b.

## 4. Discussion

An important feature of the Ag-La alloying system is the negligible solubility of lanthanum in silver (0.06 wt.% La at 1065 K [28]). Consequently, low alloyed Ag-La alloys (<12.5 wt.% La) exhibit in the as-cast state a coarse two-phase microstructure consisting of primary α_Ag_ dendrites and eutectic in the interdendritic spaces, as shown in Figure 12a. The eutectic consisted of lamellae α_Ag_ and intermetallic Ag_5_La, where direct oxidation of the intermetallic Ag_5_La forms a high inhomogeneous oxide distribution in such a microstructure after internal oxidation (Figure 12b). To attain a fine dispersion of the oxide particles in the Ag-La alloys, the coarse intermetallic Ag_5_La segregate/aggregate has to be refined to a much finer scale. Therefore, in the first part of this work, we checked the suitability of the rapid solidification technique for microstructural refinement and reducing the intermetallic segregate/aggregate spacing in Ag–2 wt.% La alloy to the submicron size, which would give the appropriate and finely dispersed oxide particles at the subsequent internal oxidation process.

The evolution of the microstructure and, consequently, the degree of microsegregation, depend on the solidification conditions, which determine the local conditions at the solid/liquid interface. In the case of rapidly solidified Ag–2 wt.% La ribbons, the nucleation of the solid phase started in the lower region of the melt, of which the surface was in contact with the quenching substrate (Figure 1a). Intensive heat release to the wheel enabled homogeneous nucleation in the highly undercooled melt, and a high concentration of fine equiaxed grains grew in the fine equiaxed zone (Figure 3). At the same time, because of the high thermal conductivity of silver, in the remaining upper region of the melt, high enough undercooling for homogeneous nucleation was also reached. However, in the upper part of the melt, the lower degree of undercooling resulted in fewer nuclei and, consequently, a much bigger grain size (Figure 4). Alternatively, the nucleation of grains in the upper region of the melt could also be started by detachment of the already existing dendrite arms in the growing FEG zone due to convection (the melt intrusion into the mushy zone). In any case, the nucleation and growth of equiaxed grains in the upper region of the melt began before the competition growth of the crystals, nucleated in the fine equiaxed zone, into liquid resulted in the formation of a positive temperature gradient and transition from the equiaxed to columnar grains.

Examination of the ribbon’s lower surface (the wheel surface in Figure 2a) revealed the presence of dimples, which also affected the evolution of the microstructure during solidification, as shown in Figure 1a. Air gaps prevent continuous contact between the melt and the wheel, and, consequently, restrain heat transfer during the cooling. This may have an influence on undercooling in the near surroundings, and result in less nucleation sides that can be associated with thinness or an uneven FEG zone. Therefore, in some regions, fine equiaxed grains decreased faster by moving from the wheel surface to the core. Additionally, where the FEG zone is thin, the upper grains in the CEG zone seem to be even more coarse (Figure 2a).

The main goal of the rapid solidification was to refine the intermetallic Ag_5_La particles to a size enabling successful IO, with fine dispersed nanosized oxide particles of the alloying element being obtained. Firstly, no metastable phase was obtained by rapid solidification, other than the extended face-centred cubic solid solution. In the ribbons, this phase was formed both in the FEG zone and at the beginning of the CEG zone, as well. This indicates the region where the interface growth rate was higher than the critical value needed for segregation-free solidification. The small size of this region (<20 µm) confirms the theory that, in alloys with a large freezing range, it is difficult to obtain a segregation-free solid [32]. A few nanosized particles of the intermetallic Ag_5_La particles inside the α_Ag_ grains were also visible in the FEG zone. We believe that these particles were formed by precipitation from the supersaturated solid solution. The diffraction patterns obtained from the larger particles could not be indexed conclusively to any of the Ag-La intermetallics reported in the literature. The composition of the intermetallic Ag_5_La particles was also analysed by EDX, and was found to be close to the structure of Ag_5_La. As shown in the results in Figure 3, the coarsest intermetallic Ag_5_La particles were situated on the grain boundaries in the rapidly solidified ribbons. This was expected, due to the fact that in globular solidification, the last melt in the interdendritic space where the boundaries between the equiaxed grains were formed contained the highest concentration of alloying elements.

In our study, the application of the rapid solidification technique as a means for microstructural refinement and reduction of the intermetallic segregate size in Ag–2 wt.% La alloy, resulting in a fine-grained microstructure where submicron-sized particles of the intermetallic phase Ag_5_La were dispersed in the noble matrix α_Ag_. According to the accepted concept of internal oxidation in two-phase alloys, the oxidation behaviour of these alloys depends critically on the solubility and diffusivity of the most reactive alloying element. The size, shape, volume fraction and type of spatial distribution of the two phases are also important factors, which determine the mechanism of the internal oxidation process. If the alloying system is composed of two components (A-B), where the alloying element B is much less noble than the base metal A, the selective oxidation of the alloy component B mostly occurs. However, in our case, the silver matrix is noble phase, and in the intermetallic Ag_5_La phase, the lanthanum is subjected to selective oxidation. Formally, in such types of two-phase alloys, the internal oxidation process can proceed in one of three different ways, or in a combination thereof:Direct “in situ” oxidation of intermetallic particles;Oxidation of the alloying element from the solid solution;Dissolution of the small intermetallic particles ahead of the Internal Oxidation Front (IOF) and oxidation of dissolved atoms of the alloying element from the solid solution.

(i) In situ, or diffusionless internal oxidation, is defined as the process where the solubility and diffusivity of B in A are very small [30,31,32]. Without any appreciable diffusion of element B, as well as fast simultaneous inward penetration of oxygen, the B present in the second-phase particles is transformed into the corresponding oxide B_m_O_n_ directly at its original location.

(ii) Conversely, if the solubility and diffusivity of B in A are high, the process of internal oxidation is denoted as classical or diffusive [30,31,32]. The internal oxidation of B will have the same basic features as the corresponding process for single-phase alloys. The most-reactive component B will be supplied to the front of internal oxidation by means of diffusion through the alloy, and the process will produce a uniform distribution of oxide particles in the noble-component matrix α_A_. The spatial distribution of oxide particles formed will not necessarily be related to the spatial distribution of the intermetallic particles in the starting alloy.

(iii) A special feature of IO in two-phase alloys with low solubility and diffusivity of B in A and small submicron-sized second-phase particles is the occurrence of a homogeneous single-phase layer of α_A_ ahead of the IOF. Its occurrence depends on the kinetics of dissolution of the intermetallic particles in front of the IOF and the rate of penetration of the IOF. At a fixed temperature, the kinetics of dissolution are determined by the saturation deficit of reactive component B in the matrix, and the rate of oxidation-front penetration decreases with the depth of IO, because of the lowering of the oxygen concentration gradient. The slower the IOF penetrates, the more time remains for dissolution of the intermetallic particles, and, at some distance, where the oxidation rate decreases enough with respect to the dissolution kinetics of the intermetallic particles, the single-phase layer of α_A_ is formed [30,31,32].

Applying the criterion of the solid solubility of lanthanum in silver in order to predict the mode of internal oxidation of the selected Ag-La alloy leads us to the conclusion that the dilute Ag-La alloy in the form of rapidly solidified ribbons should oxidise internally through the diffusionless or “in situ” mode. Contrary to this, the morphology and size of the oxide particles in the oxidised ribbons revealed three different types of oxide particles: (I) elongated, in a few hundreds of nm size range La_2_O_3_ particles of similar shape and spatial distribution as the Ag_5_La intermetallic particles in the original alloy (Figure 6); (II) globular, in a few tens of nm size range La_2_O_3_ particles in the interior of the grains (Figure 6); and (III) super fine, in a few nm size range oxide precipitates quite homogeneously distributed in the silver matrix, with higher density in the vicinity of the globular La_2_O_3_ particles (Figure 5b).

The lanthanum content in the silver matrix surrounding the tiny oxide precipitates in the FEG was found to be practically zero, while the lanthanum content in the matrix of the non oxidised region of the alloy was found to approach to the maximum solubility limit of about 0.06 wt.% of La. This type of concentration profile of element B (the most reactive element), ranging from zero at the IOF to the solid solubility limit in the unoxidised alloy, together with the typical morphology and spatial distribution of the tiny oxide precipitates, is characteristic of classical internal oxidation as found in single-phase alloys. Therefore, super fine, tiny oxide particles can arise, either by precipitation of the oxides from a supersaturated solid solution, or by a dissolution/reprecipitation mechanism (dissolution of finer intermetallic particles ahead of the IOF and oxidation of La from the solid solution). According to the fact that the velocity of decomposition of the supersaturated solid solutions is much higher than the velocity of internal oxidation, most of these particles were probably formed by the second mechanism. The reaction of lanthanum with oxygen at the IOF, as well as precipitation of the oxide product from the α_Ag_ matrix, creates a concentration gradient and diffusion of solute lanthanum towards the interface. Consequently, the region ahead of the IOF is depleted by lanthanum, and the finely dispersed intermetallic Ag_5_La particles start to dissolve. However, if the intermetallic Ag_5_La particles are too coarse for complete dissolution, the oxidation of the non-dissolved rest of the intermetallic Ag_5_La particles continues with direct oxidation in the internal oxidation zone after the IOF’s passing, and, finally, completes the direct oxidation, resulting in roundish, coarser oxides about a few tens nm in size in the interior of the grains. On the other hand, the oxides at the grain boundaries were elongated, and their distribution corresponded to those of the coarser intermetallic Ag_5_La particles prior to oxidation. Additionally, the vicinities of these oxides were without any fine, tiny oxide particles. All these facts indicate that intermetallic Ag_5_La particles at the grain boundaries were probably oxidised only by “in situ” direct oxidation, without their dissolution ahead of the IOF.

Additionally, with metallographic quantitative analysis it was confirmed that the size of the oxide La_2_O_3_ particles obtained by direct oxidation of the intermetallic Ag_5_La particles in the rapidly solidified ribbons was much smaller than the size of the intermetallic Ag_5_La particles prior to oxidation. The average diameter of the oxide La_2_O_3_ particle was 99 nm, while the average diameter of the primary intermetallic Ag_5_La particle was 211 nm. In fact, all intermetallic particles that were oxidised by the mechanism of direct oxidation were of much smaller size after oxidation. In the case of the intermetallic particles that were, in the first step of the oxidation process, partially dissolved ahead of the IOF, the reduction of the particles’ sizes after oxidation and the change in the morphology can definitely be attributed partially to the process of dissolution of the intermetallic particles ahead of the IOF.

However, because of the high difference in the size of the starting intermetallic particles and the oxides, an additional cause for decreasing the size of the directly oxidised intermetallic particles must be hidden in the mechanism of oxidation of the intermetallic Ag_5_La compound. Therefore, the mechanism of oxidation of an intermetallic Ag_5_La compound has been studied supplementarily in a hypereutectic Ag-14 wt.% La alloy in the as-cast state, and is discussed here to reveal the additional source of decreasing the size of the oxide particles.

Microstructural analysis after oxidation revealed a considerable rearrangement of the Ag and La atoms at the reaction front. The oxidation of La from the intermetallic Ag_5_La compound induced the depletion of La atoms ahead of the IOF, and therefore caused diffusion of the La atoms toward the IOF from the unoxidised region. At the same time, a high excess of Ag atoms was formed at the reaction front, named Ag segregate. This contributed to diffusion of the Ag atoms in the opposite direction from the IOF. In fact, Ag diffuses to the rest of the intermetallic Ag_5_La particles, and also, laterally, the needles of La_2_O_3_ are formed and the Ag segregate stuck in the inter-needle space, as shown in Figure 13. The formation of oxide needles can be observed trough the volume of prior Ag_5_La dendrite (Figure 13b), as well as in the prior intermetallic Ag_5_La lamellae (Figure 13c). As the reaction front moves through the dendrite, the segregated Ag thrusts forward before the IOF. At the edge of the preceding intermetallic Ag_5_La dendrite, this results in the formation of a crescent of segregated Ag. More precisely, a crescent is formed on the opposite side of dendrite from where the oxidation has begun, as can be seen on the top of the dendrite in Figure 13a.

The described oxidation proceeding in the intermetallic Ag_5_La is valid for the coarse intermetallic particles. This results in the morphology denoted here as “oxidation morphology of big Ag_5_La particles”, which is presented schematically in Figure 14. A comparable oxidation process can also be found in the coarse Ag_5_La lamella of the eutectic, where the Ag_5_La lamellae are about 4 µm or wider. Consequently, during internal oxidation, they are transformed into oxide needles and segregated Ag. These segregated Ag are bordered by α_Ag_ lamellae, and their boundary cannot be observed exactly, as can be seen clearly in Figure 13c.

Actually, when considering the thickening of the oxide needles (from Figure 13) in the reaction zone of the IOF in the region of the oxidised coarse intermetallic particles, the maximum lateral diffusion path (x) of La atoms during oxidation at 973 K can be determined by measuring the mutual distance (2x) between the individual oxide needles (Figure 15). With the measurements in the oxidised Ag_5_La dendrites and oxidised Ag_5_La eutectic lamellae, the mean diffusion distance (x) was found to be about 0.75 µm.

Taking into account the mean thickness of the oxide needles (d = 0.68 µm), the periodicity of the growing oxide needles (ʎ) was calculated as:ʎ = 2x + d(1)
and the value ʎ = 2.18 µm was obtained. This finding contributes to an important factor in determining the oxidation morphology as a function of the size of the intermetallic Ag_5_La particles. Namely, it is expected that oxidation of intermetallic Ag_5_La particles larger than 2 µm will result at 973 K in “oxidation morphology of big Ag_5_La particles”, while intermetallic particles smaller than 2 µm will be oxidised into only one oxide aggregate, thus forming the morphology presented schematically in Figure 16. We denoted this morphology as “oxidation morphology of small Ag_5_La particles”. In the case of small intermetallic Ag_5_La particles, the oxygen atoms again prefer to react chemically at the site of the La-rich region on the α_Ag_/Ag_5_La interface (particle border). The selective oxidation of La from the intermetallic Ag_5_La compound results again in depletion of La and excess of Ag ahead of the oxidation front. Consequently, diffusion of the atoms of both components changes the composition in the unoxidised rest of the intermetallic Ag_5_La compound continuously. However, a short enough diffusion distance from the edge to the core of the small intermetallic particles enables the continuous growth of only one oxide aggregate (Figure 6), which is, therefore, coated laterally and on the top with the segregated Ag. Therefore, after completed oxidation of small intermetallic Ag_5_La particles (smaller than 2 µm), the volume of particles is transformed into La_2_O_3_ oxide aggregate and a pure segregated Ag coat, as is shown in Figure 16c. In fact, this oxidation morphology, denoted here as “oxidation morphology of small Ag_5_La particles”, presents the “missing” additional source of the huge decrease in the size of the oxide particles which were formed by direct oxidation of the intermetallic Ag_5_La particles. Therefore, in the case of our rapidly solidified ribbons where the size of intermetallic Ag_5_La particles was deeply in the submicron region (Figure 4), it is quite obvious that, after internal oxidation, the volume of the oxide particles will be much smaller than the volume of the original intermetallic Ag_5_La particles.

For the second case, where direct oxidation of enough small intermetallic particles results in the so-called “oxidation morphology of small Ag_5_La particles”, the relationship between the volume of the La_2_O_3_ oxide and the volume of the original intermetallic Ag_5_La particles can also be obtained theoretically by the ratio of the volume of the elementary cell of a metal oxide to the volume of the elementary cell of the corresponding intermetallic compound (the Pilling-Bedworth ratio)—as presented in Figure 16. In principle, a hexagonal unit cell of Ag_5_La (Pearson Symbol (PS): hP12; a = 0.5569 nm; c = 0.9077 nm), which contains 10 atoms of Ag and 2 atoms of La, is, during oxidation, converted into a hexagonal unit cell of La_2_O_3_ (PS: hP5; a = 0.3938 nm; c = 0.6136 nm), which contains two atoms of La and three atoms of O, and into a cubic cell of Ag (PS: cF4; a = 0,40853 nm) with four atoms in a unit cell. More precisely, two-unit cells of Ag_5_La (V_2Ag5La_ = 0.487592 nm^3^) transform during oxidation into two-unit cells of La_2_O_3_ (V_2La2O3_ = 0.164814 nm^3^) and five-unit cells of Ag (V_5Ag_ = 0.340910 nm^3^). The comparison of the volume of La_2_O_3_ and the volume of the Ag_5_La intermetallic compound (V_La2O3_/V_Ag5La_ = 0.3380) reveals that the volume of the oxide particles which arise during oxidation of small particles of the Ag_5_La intermetallic compound present only 33.8% of the volume of the original intermetallic compound. Therefore, decreasing the size of the intermetallic compound in the starting microstructure prior to internal oxidation to the values that enable the “oxidation morphology of small Ag_5_La particles” will, essentially, decrease the final size of the La_2_O_3_ oxide particles.

If both particles (the intermetallic and the oxides) are considered to be spherical, the obtained ratio of the volumes (V_La2O3_/V_Ag5La_ = 0.3380) corresponds to the value of 0.696 for the particles’ radii ratio (R_La2O3_/R_Ag5La_). On the other hand, the ratio of the radii obtained by metallographic quantitative measurements was mostly lower (0.35–0.60), and only for the particles at the grain boundaries in the top layer of the ribbons was the obtained ratio 0.70. Hence, it is obvious that the majority of intermetallic particles dissolve ahead of the IOF before they start to oxidise directly.

Finally, the sum of the volumes of the oxidation products is greater than the original volume of the intermetallic compound by more than 3% ([V_La2O3_ + V_Ag_]/V_Ag5La_ = 1.0371). Therefore, direct oxidation of small homogeneously distributed Ag_5_La intermetallic particles is accompanied by a relatively large volume expansion, and, consequently, the compression stress zone in the vicinity of the oxides also leads to hardening of the surrounded matrix.

## 5. Conclusions

The oxidation behaviour of microstructurally highly metastable Ag-La alloy was investigated in this paper. The following conclusions were drawn:The cross-section of the RS Ag-La ribbons consists of two zones: FEG and CEG, where the diameter of the grains in the FEG zone extended from 150 nm to 900 nm, and in the CEG zone from 1 µm to 5 µm.The RS process was used successfully to achieve finely dispersed intermetallic Ag_5_La particles of small sizes, making it possible to attain fine dispersion of nanosized La_2_O_3_.Three mechanisms of internal oxidation were identified: (i) the oxidation of La from the solid solution; (ii) partial dissolution of finer Ag_5_La particles before the internal oxidation front and oxidation of La from the solid solution; (iii) direct oxidation of coarser Ag_5_La intermetallic particles.To attain the nanosized oxide particles in the case of direct oxidation of intermetallic particles, two conditions must be fulfilled: (i) the intermetallic phase must be composed of noble and high reactive elements so that oxidation proceeds with selective oxidation of only the high reactive element; (ii) the size of the intermetallic particles must be small enough to enable the formation of only one oxide aggregate.

## Figures and Tables

**Figure 1 materials-15-02295-f001:**
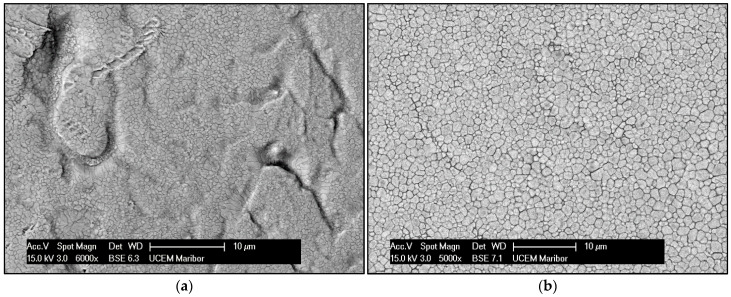
SEM analysis of a rapidly solidified ribbon: (**a**) lower side (wheel surface) of a ribbon with dimples in the surface; (**b**) upper side (free surface) of a ribbon.

**Figure 2 materials-15-02295-f002:**
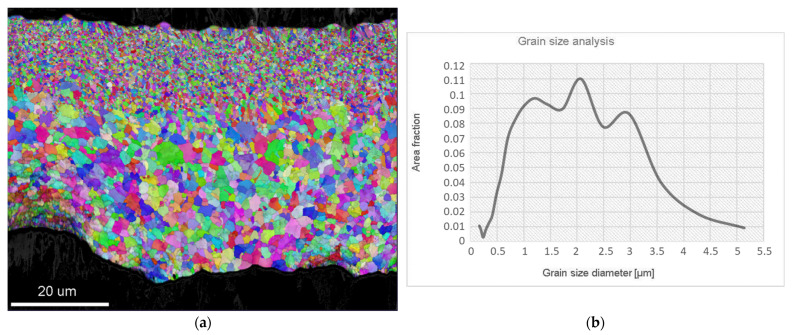
Grain size analysis in a transverse cross-section of the rapidly solidified ribbons: (**a**) EBSD mapping; (**b**) graph of grain diameter vs. area fraction.

**Figure 3 materials-15-02295-f003:**
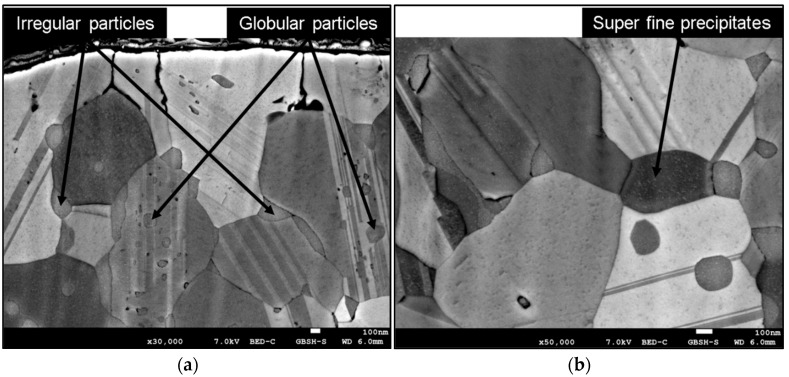
SEM micrograph of a rapidly solidified microstructure in the FEG zone: (**a**) overview of the microstructure; (**b**) detail of the FEG zone.

**Figure 4 materials-15-02295-f004:**
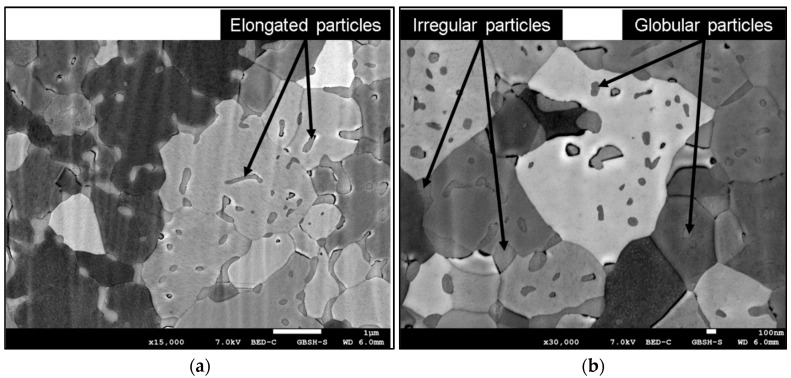
SEM micrograph of a rapidly solidified microstructure in the CEG zone: (**a**) overview of microstructure; (**b**) distribution of intermetallic particles inside the grains.

**Figure 5 materials-15-02295-f005:**
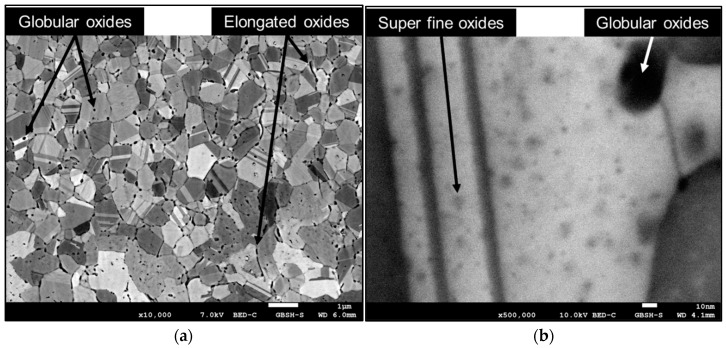
SEM micrograph of the internally oxidised microstructure from the bottom layer of the ribbon (FEG zone after RS): (**a**) overview of a bottom layer; (**b**) detail from the bottom layer with super fine oxide particles in the αAg grain.

**Figure 6 materials-15-02295-f006:**
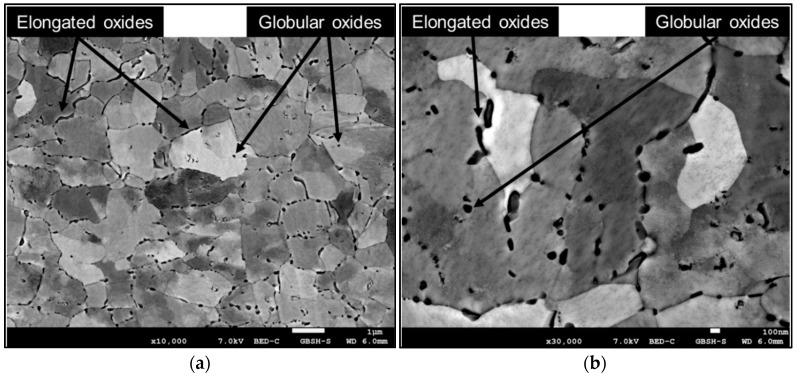
SEM micrograph of the internally oxidised microstructure from the top layer of the ribbon (CEG zone after RS): (**a**) overview of the top layer; (**b**) detail from the top layer.

**Figure 7 materials-15-02295-f007:**
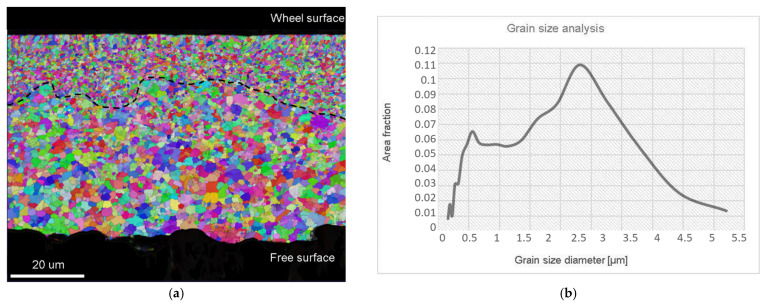
Grain size analysis of a transverse cross-section of the internally oxidised microstructure: (**a**) EBDS mapping at magnification 1500×; (**b**) graph of grains’ diameter vs. area fraction.

**Figure 8 materials-15-02295-f008:**
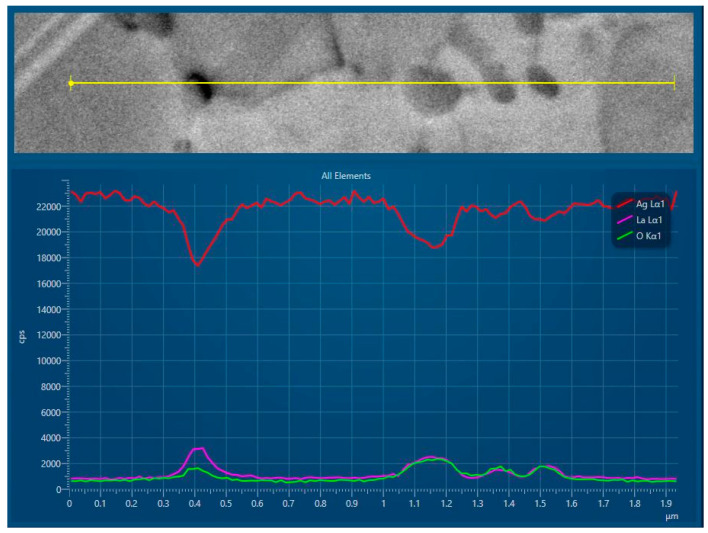
Line microchemical analysis of an internally oxidised ribbon.

**Figure 9 materials-15-02295-f009:**
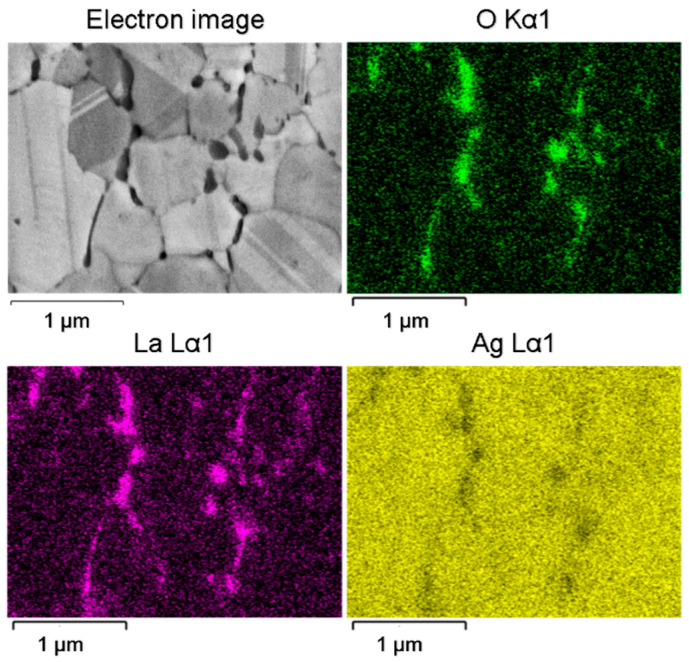
Micrograph of the chemical mapping of an internally oxidised microstructure with detection of the elements Ag, O and La.

**Figure 10 materials-15-02295-f010:**
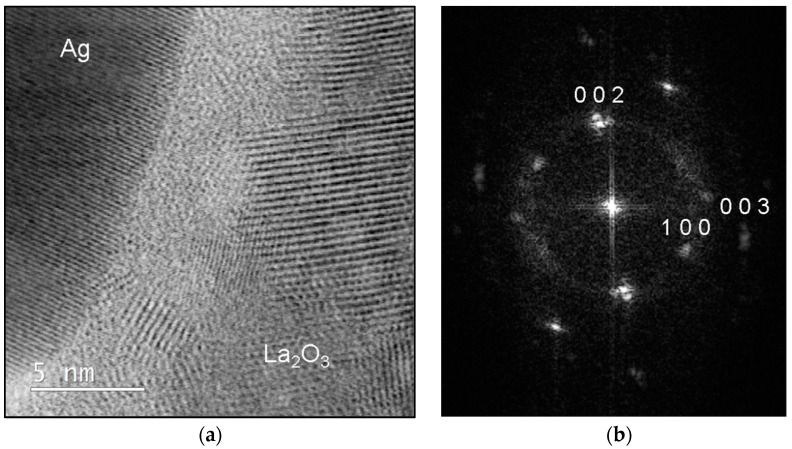
STEM analysis of the internally oxidised ribbons: (**a**) bright-field image of the interface be-tween the αAg and La_2_O_3_; (**b**) corresponding FFT pattern from the bright-field image of the La_2_O_3_ oxide.

**Figure 11 materials-15-02295-f011:**
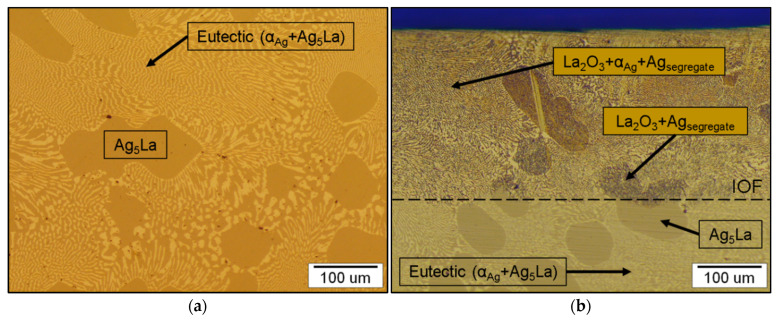
Microstructure of the cast hypereutectic composition of Ag-14 wt.% La: (**a**) as cast; (**b**) after partial internal oxidation, with a visible Internal Oxidation Front (IOF).

**Figure 12 materials-15-02295-f012:**
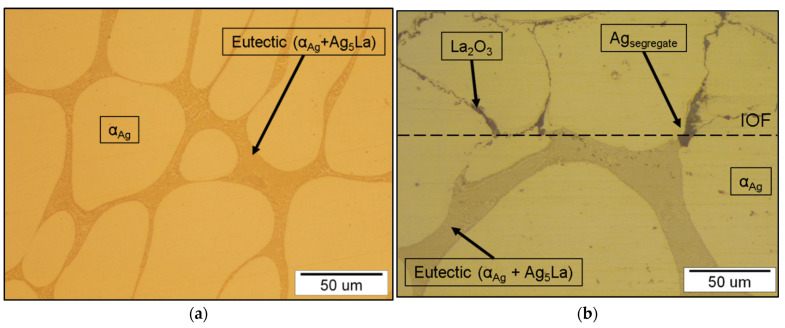
Microstructure of the Ag–2 wt.% La alloy: (**a**) as-cast and (**b**) transition between the non-oxidised (lower side) and internally oxidised (upper side) cast microstructure.

**Figure 13 materials-15-02295-f013:**
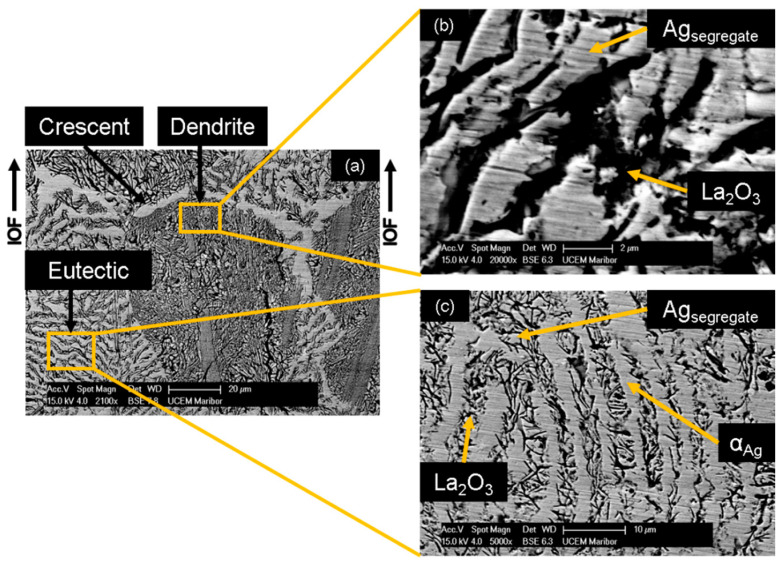
SEM micrograph of a hypereutectic microstructure: (**a**) overview of the completely oxidised microstructure; (**b**) detailed micrograph from the oxidised dendrite; (**c**) detail from the oxidised lamellae in the eutectic.

**Figure 14 materials-15-02295-f014:**
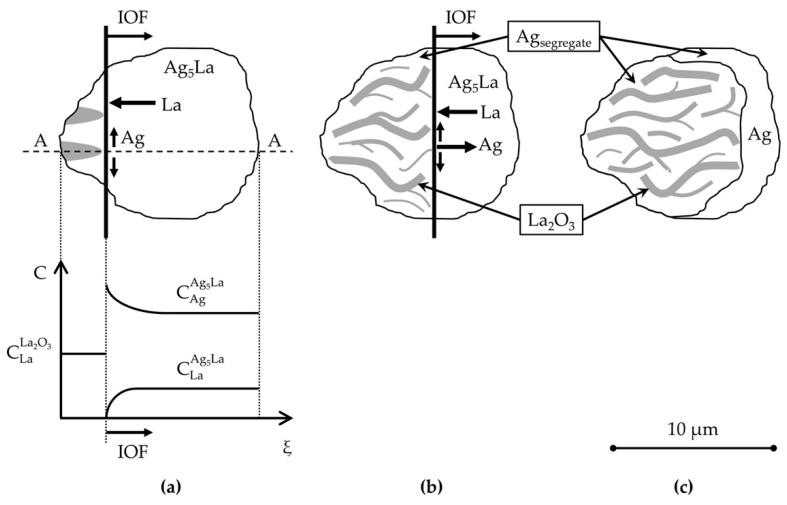
The model of internal oxidation of a “big Ag_5_La particle” in the hypereutectic microstructure with moving internal oxidation front: (**a**) beginning of oxidation with a concentration profile of Ag and La; (**b**) intermediate stage; (**c**) completed oxidation with a crescent on the last oxidised side.

**Figure 15 materials-15-02295-f015:**
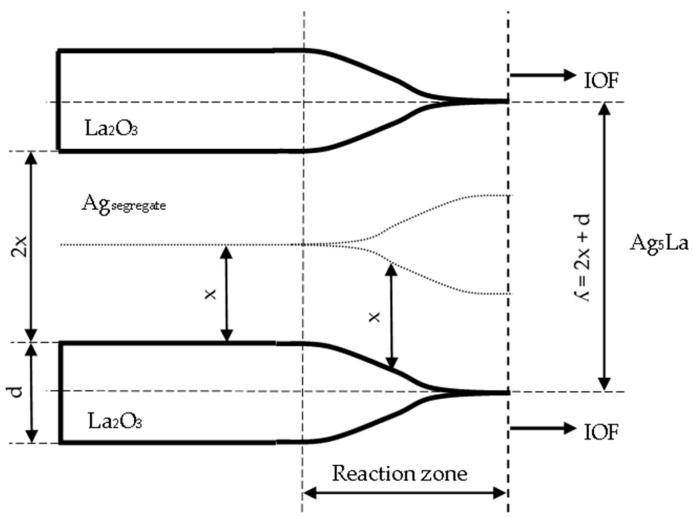
The model of oxide thickening in the reaction zone of the “Big Ag_5_La particles” at the internal oxidation front.

**Figure 16 materials-15-02295-f016:**
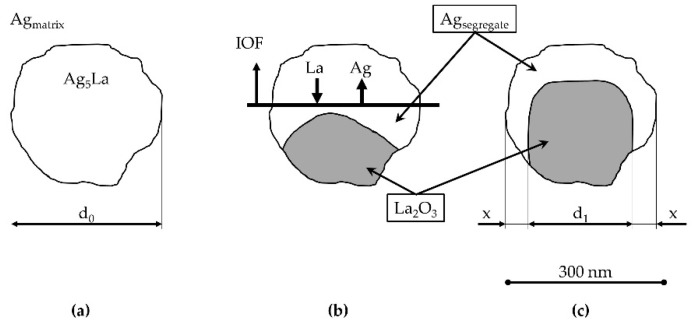
The oxidation model of “Small Ag_5_La Particles” in the hypoeutectic microstructure with an upward moving internal oxidation front: (**a**) submicron intermetallic particle before oxidation; (**b**) re-arrangement of atoms during oxidation; (**c**) separated area with oxide and segregated Ag.

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
