# Peer review of "Oxidation Behaviour of Microstructurally Highly Metastable Ag-La Alloy"

_materials, 2022, doi:10.3390/ma15062295_

Round 1

Reviewer 1 Report

An interesting work for internal oxidation of Ag-La alloy prepared by rapid solidification is introduced and different oxidation mechanisms are discussed. However, the novelty is not enough presented and the discussion part should be reconsidered. The detailed suggestions are listed below.

  1. There seems no  effect of grain size on IO results, so the presentation of grain size comparison can be shortened;
  2. compared between Fig.5 and Fig.6, obvious difference is amount of super fine oxides in the order of few nm in the interior of the grains for FEG zone, the mechanism for this is not discussed and enough analysis for this difference is lacking;
  3. The oxidation mechanism is discussed mainly based on the observation of a hypereutectic alloy, is it applicable of the deduced equation 1 for Ag-2La alloy especially prepared by RS process?
  4. In the abstract, " The fine dispersion of La2O3 nanoparticles in the silver matrix improves the mechanical, as well as functional properties,greatly“ is not studied in this work and suggest to delet;
  5. In the abstract, the description seems not correct: " (ii) Dissolution of fine Ag5La before the internal oxidation front and oxidation of 22
    La from the solid solution; (iii) Direct oxidation of coarser Ag5La intermetallic particles."
  6. The conclusion 6 needs more support results;
  7. there are many mistakes in figure number in the context, for example, the fig.13 should be fig.14 discussed in line 710-718; fig.14  should be fig.15 in line 760; fig.13c should be fig.14c; fig.15 should be fig.16 in 794-795; fig.16 should be fig.17 in line 842-845;  

Author Response

Reviewer 1

  1. There seems no  effect of grain size on IO results, so the presentation of grain size comparison can be shortened;
  2. compared between Fig.5 and Fig.6, obvious difference is amount of super fine oxides in the order of few nm in the interior of the grains for FEG zone, the mechanism for this is not discussed and enough analysis for this difference is lacking;
  3. The oxidation mechanism is discussed mainly based on the observation of a hypereutectic alloy, is it applicable of the deduced equation 1 for Ag-2La alloy especially prepared by RS process?
  4. In the abstract, " The fine dispersion of La2O3 nanoparticles in the silver matrix improves the mechanical, as well as functional properties,greatly“ is not studied in this work and suggest to delet;
  5. In the abstract, the description seems not correct: "(ii) Dissolution of fine Ag5La before the internal oxidation front and oxidation of 22
    La from the solid solution; (iii) Direct oxidation of coarser Ag5La intermetallic particles."
  6. The conclusion 6 needs more support results;
  7. there are many mistakes in figure number in the context, for example, the fig.13 should be fig.14 discussed in line 710-718; fig.14  should be fig.15 in line 760; fig.13c should be fig.14c; fig.15 should be fig.16 in 794-795; fig.16 should be fig.17 in line 842-845;  

Author's explanation and answers:

Ad 1.) As is well known, internal oxidation is a diffusion-controlled process involving a selective reaction of a less noble solute or second phase particles, with oxygen diffusing in from the surface. At the same time, coarsening of the second phase particles (Ostwald ripening), as well as decomposition of the supersaturated solid solution (because of the highly metastable rapidly solidified microstructure) occur ahead of the internal oxidation front (both processes are also diffusion controlled). Therefore, the kinetic of the diffusion of oxygen atoms determines the mechanism of internal oxidation, and, consequently, the nature and the size of the oxide particles. There are two mechanisms that determine the kinetic of diffusion, namely, bulk and grain boundary diffusion. However, the main mechanism responsible for faster diffusivity in rapidly solidified ribbons is believed to be the grain boundary diffusion. Therefore, the size of grains influences the internal oxidation significantly. For instance, in the zone of Fine Equiaxed Grains (FEG) the rate of penetration of the internal oxidation front is much faster than in the zone of Coarse Equiaxed Grains (CEG). Consequently, in the FEG many more oxide particles are formed by oxidation of the alloying element from the supersaturated solid solution (therefore, super fine oxide particles are present in the FEG but not in the CEG), and also in the FEG there is much less time for Ostwald ripening of the intermetallic particles ahead of the Internal oxidation front. Therefore, detailed analysis of the size of grains helps us to understand the mechanism of internal oxidation and the achieved microstructure.

To emphasise the importance of grain size on the kinetic of diffusion, and, consequently, on the mechanism of internal oxidation, the following sentence has been added: “Since the main mechanism responsible for faster diffusivity in rapidly solidified ribbons is the grain boundary diffusion, the size of grains influences the internal oxidation significantly. “

Ad 2.) The difference in the amount of super fine oxides is already explained in the answer to the first question. Let me repeat. In the zone of Fine Equiaxed Grains (FEG), the rate of penetration of the internal oxidation front is much faster than in the zone of Coarse Equiaxed Grains (CEG). Consequently, in the FEG, many more oxide particles are formed by oxidation of the alloying element from the supersaturated solid solution. Additionally, because the intermetallic Ag5La particles are much finer in the FEG (greater curvature), the process of partial dissolution ahead of the internal oxidation front and oxidation of La from the solid solution (which resulted again in super fine oxides), is much more intensive than in the CEG.

These mechanisms and analysis are presented in the Discussion (lines 663 – 688). For even more clarity, the name of the zone to which the discussion relates (“in the FEG“) was added at the beginning of this paragraph.

Ad 3.) We believe so. In fact, the chemical composition and the structure of the intermetallic Ag5La phase is the same for both the conventionally cast hypereutectic Ag – 14 wt.% La alloy and the rapidly solidified Ag – 2 wt.% La. The mechanism of direct oxidation of the intermetallic phase takes place the same in both compositions: The oxygen atoms react chemically at the site of the La-rich region on the αAg/Ag5La interface, and the selective oxidation of La results in depletion of La and excess of Ag ahead of the oxidation front. Consequently, diffusion of the atoms of both components changes the composition in the unoxidised rest of the intermetallic Ag5La compound continuously. The difference is only in the morphology of the oxide formed, which is explained in more detail in the Discussion.

Ad 4.) We agree with your proposal and the sentence has been removed.

Ad 5.) We agree with your finding. The sentence has been changed: “(ii) Partial dissolution of the finer Ag5La particles before the internal oxidation front, and oxidation of La from the solid solution; (iii) Direct oxidation of the coarser Ag5La intermetallic particles.“

Ad 6.) The mechanism of direct oxidation of intermetallic Ag5La phase (the intermetallic composed of noble (Ag) and high reactive element (La)), definitely consists of selective oxidation of La from the intermetallic Ag5La compound, and, consequently, results in depletion of La and excess of Ag ahead of the oxidation front. Diffusion of the atoms of both components changes the composition in the unoxidised rest of the intermetallic Ag5La compound and influences the morphology of the oxidised region. In the case of coarse intermetallic particles (primary dendrites of the Ag5La intermetallic phase in the microstructure of the hypereutectic Ag-14 wt. %La alloy) the oxidation results in the needles of La2O3 and segregated Ag. This morphology has been confirmed in several experiments. On the other hand, with the study of internal oxidation in rapidly solidified ribbons, where the intermetallic particles smaller than 1 mm oxidise into only one oxide aggregate, the model of oxidation morphology was confirmed as valid for Small Ag5La particles. The lateral and frontal segregation of the Ag are the cause that the volume and the size of the oxide particles are much smaller than the volume and the size of the original intermetallic Ag5La particles.

We believe that conclusion 6 was completely confirmed with the experimental results, and it gives very important instruction on how to attain nanosized oxide particles in the case of direct oxidation of intermetallic particles.

For more clarity, the Conclusion 6 has been modified. “To attain the nanosized oxide particles in the case of direct oxidation of intermetallic particles, two conditions must be fulfilled: (i) The intermetallic phase must be composed of noble and high reactive elements, so that oxidation proceeds with selective oxidation of only the high reactive element; (ii) The size of the intermetallic particles must be small enough to enable the formation of only one oxide aggregate. “

Ad 7.) Yes, unfortunately we did a series of mislabelling of images in the text. We have fixed the mistakes.

Reviewer 2 Report

The paper reports the oxidation behaviour of microstructurally highly metastable Ag‐La alloy. The authors prepared the alloys by rapid solidification. The overall quality of the manuscript is good. However, the following revisions are required, before considering the acceptance for publication in materials journal:

Q1) Why does rapid solidification enable high microstructural refinement and provided a suitable starting microstructure for the subsequent internal oxidation?

Q2) How to determine the oxidation is from rapid solidification or heat treatment?

Q3) As the author also mentioned, high‐temperature oxidation is mostly undesirable, and they often cause the corrosive failure of engineering components so it is better to do corrosion test for each step.

Q4) The author can introduce any specific application for prepared alloy?

Q5) In addition of STEM analysis, to measure elemental composition as well as the chemical and electronic state of the atoms within the alloy better to do XPS test

Q6) In the last paragraph of the introduction and before starting materials and method, the author needs to write the statement of purpose of this research and what will investigate in this paper.

Author Response

Reviewer 2

The paper reports the oxidation behaviour of microstructurally highly metastable Ag‐La alloy. The authors prepared the alloys by rapid solidification. The overall quality of the manuscript is good. However, the following revisions are required, before considering the acceptance for publication in materials journal:

Q1) Why does rapid solidification enable high microstructural refinement and provided a suitable starting microstructure for the subsequent internal oxidation?

Q2) How to determine the oxidation is from rapid solidification or heat treatment?

Q3) As the author also mentioned, high‐temperature oxidation is mostly undesirable, and they often cause the corrosive failure of engineering components so it is better to do corrosion test for each step.

Q4) The author can introduce any specific application for prepared alloy?

Q5) In addition of STEM analysis, to measure elemental composition as well as the chemical and electronic state of the atoms within the alloy better to do XPS test

Q6) In the last paragraph of the introduction and before starting materials and method, the author needs to write the statement of purpose of this research and what will investigate in this paper.

Author's explanation and answers:

Ad Q1) Rapid solidification has already been well known for a long time as a means for refinement of the microstructure of metallic based materials. Generally, rapid solidification is considered to be a “high non-equilibrium“ process, in which the degree of departure from equilibrium is determined by the degree of undercooling that determines the rate of solidification (the growth rate of the solid/liquid interface). High undercooling is either obtained by rapid cooling (melt spinning, atomising, ...) or, in slow cooling, by the reduction of heterogeneous nucleation (levitation, flux immersion, ...). When the growth rate of a solid/liquid interface exceeds the rate of diffusion of the solute atoms in the liquid phase, significant loss of interfacial equilibrium can occur. Phase diagrams fail to give the temperature and compositions at the interface, and, consequently, the chemical potentials of the atoms in the solid and liquid at the interface are no longer  equal. In such conditions, the rapid solidification of a solid solution system results in the solid having a composition close to that of the liquid composition, because there is insuficient time for the solute to redistribute in the liquid during the solidification process – a supersaturated solid solution. Similarly, in eutectic systems, the solute trapping also results in a metastable extension of the region of solid solution and/or fine dispersion of the second phase particles. However, if the composition of the alloy is near the eutectic composition, the rapid solidification can result in celular morphology of one of the phases, while the second phase grows/precipitates in the intercellular space. Aditionally, in the case of eutectic alloys, it may be possible to undercool the whole system to a temperature below the glass transition temperature, Tg, whereupon a metastable amorphous structure results. Therefore, it is obvious that the morpfology of the phases is dependent on the solidification front velocity, chemical diffusion and thermal diffusion.

In the case of our alloy, which conventionally solidifies into dendrites of primary aAg and coarse eutectic (aAg + Ag5La) in the interdendritic space with the grain size of a few tens of mm, the rapid solidification using melt spinning technology resulted in fine, mostly submicron grains of supersaturated aAg solid solution and a fine dispersion of the Ag5La intermetallic phase. While all these phenomena (degree of supersaturation, portion of grain boundaries – the size of the grains, the size of the intermetallic Ag5La particles) influence and determine the size of the final La2O3 oxides, the rapid solidification provided a suitable starting microstructure for internal oxidation.

Ad Q2) The experiments of rapid solidification were conducted in an atmosphere of pure argon (5.0).  Therefore, no oxidation occurred during rapid solidification and the oxides have been formed in the controlled Internal oxidation procedure.

Ad Q3) It is true that high-temperature oxidation is mostly undesirable. Contrary to this, the controlled internal oxidation resulting in fine dispersion of nanosized oxide particles can be used for dispersion hardening of metals and alloys and for improvement the functional properties of the electrical contact materials.

Ad Q4) Ag-based electrical contact materials (It was mentioned in the Introduction; lines 50 - 75).

Ad Q5) XPS is a very useful method of analysis, and is used mostly as a surface analytical technique to identify the chemical elements, their concentration and oxidation states. However, it applies that the electronic states of the atoms in the intermetallic Ag5La phase do not influence the mechanism of internal oxidation and the morphology of the oxides critically. Therefore, we believe that XPS analysis is not crucial for the results of our research work, in which the aim was to attain uniformly distributed, fine, incoherent oxide particles in the Ag matrix.

Ad Q6) The purpose of our research, of which the results are reported and presented in the proposed paper (not investigated in the paper), is described at the end of the last paragraph:

“Therefore, RS performed by the Melt Spinning technique was used in our research work to investigate the influence of the starting highly metastable microstructure in the selected Ag – 2 wt.% La alloy on the IO proceedings, and the process tracing served as a tool to define the mechanism of oxidation. Additionally, the high-temperature IO in the hypereutectic Ag – 14 wt. % La was studied in the as-cast state of the alloy, to clarify the oxidation mechanism of the Ag5La intermetallic phase.”

For even more clarity the aim will be added (lines 131-133): “The objective was to identify an optimal microstructure that will lead to uniformly distributed, nanosized, incoherent oxide particles in the Ag matrix.“

Reviewer 3 Report

  1. The references should be improved and the researches of past 3 to 5 years should be added and reviewed in the manuscript.
  2. The conclusions should be refined,maybe 3 to 4 are enough and suitable. 

Author Response

Reviewer 3

  1. The references should be improved and the researches of past 3 to 5 years should be added and reviewed in the manuscript.
  2. The conclusions should be refined,maybe 3 to 4 are enough and suitable. 

Author's explanation and answers:

Ad 1.) We followed the proposal. The new references have been added [1-3].

Ad 2.) We followed the proposal. The conclusions have been refined.

Reviewer 4 Report

The paper ”Oxidation behaviour of microstructurally highly metastable Ag‐La alloy” is suitable for publication after a minor addition to the paper.

The introduction is well written and also there are lots of SEM and STEM images showing the microstructure.

In my opinion, some XRD analysis will complete the microstructure characterization and will be suitable to be publicated.

Author Response

Reviewer 4

The paper ”Oxidation behaviour of microstructurally highly metastable Ag‐La alloy” is suitable for publication after a minor addition to the paper.

The introduction is well written and also there are lots of SEM and STEM images showing the microstructure.

In my opinion, some XRD analysis will complete the microstructure characterization and will be suitable to be publicated.

Author's explanation and answers:

XRD analysis is a nice tool, and enables the identification of phases based on their diffraction pattern. In our case, the composition of intermetallic particles has been analysed very carefully by Energy-Dispersive Spectroscopy, and has been found in the region of about 18 at. %, which is very close to the composition of Ag5La. However, the exact nature of these intermetallic particles is not critical for the mechanism of the internal oxidation process, and, consequently, deeper study of the nature of these particles is not necessary. On the other hand, identification of the oxides obtained by the oxidation of intermetallic particles is very important. It has been performed by Electron Diffraction in STEM and they were confirmed as stable La2O3 oxides.

Round 2

Reviewer 1 Report

All the comments are well responded.

Reviewer 2 Report

The manuscript has been considerably improved and all the comments are answered. I recommend it to be published in the journals